# Identification of Novel Nucleocapsid Chimeric Proteins Inhibiting HIV-1 Replication

**DOI:** 10.3390/ijms232012340

**Published:** 2022-10-15

**Authors:** Hae-In Kim, Ga-Na Kim, Kyung-Lee Yu, Seong-Hyun Park, Ji Chang You

**Affiliations:** 1National Research Laboratory of Molecular Virology, Department of Pathology, School of Medicine, The Catholic University of Korea, 222 Banpo-daero, Seocho-gu, Seoul 06591, Korea; 2Graduate Program in Bio-industrial Engineering, College of Life Science and Biotechnology, The Yonsei University, 50-1 Yonsei-ro, Seodaemun-gu, Seoul 03722, Korea

**Keywords:** HIV-1, nucleocapsid, Tat, HEXIM1, transcription, packaging

## Abstract

The positive transcription elongation factor b (P-TEFb) is an essential factor that induces transcription elongation and is also negatively regulated by the cellular factor HEXIM1. Previously, the chimeric protein HEXIM1-Tat (HT) was demonstrated to inhibit human immunodeficiency virus-1 (HIV)-1 transcription. In this study, we attempted to develop an improved antiviral protein that specifically binds viral RNA (vRNA) by fusing HT to HIV-1 nucleocapsid (NC). Thus, we synthesized NC-HEXIM1-Tat (NHT) and HEXIM1-Tat-NC (HTN). NHT and HTN inhibited virus proliferation more effectively than HT, and they did not attenuate the function of HT. Notably, NHT and HTN inhibited the infectivity of the progeny virus, whereas HT had no such effect. NHT and HTN selectively and effectively interacted with vRNA and inhibited the proper packaging of the HIV-1 genome. Taken together, our results illustrated that the novel NC-fused chimeric proteins NHT and HTN display novel mechanisms of anti-HIV effects by inhibiting both HIV-1 transcription and packaging.

## 1. Introduction

Eukaryotic transcription is tightly regulated by various cellular factors. One such factor is positive transcription elongation factor b (P-TEFb), which is essential for regulating transcription initiation mediated by RNA polymerase II (RNAPII). P-TEFb is a multiprotein complex composed of cyclin-dependent kinase 9 (CDK9) as the catalytic subunit and cyclin T1 (CycT1) as the regulatory subunit [1,2,3]. P-TEFb can phosphorylate the carboxyl-terminal domain (CTD) of the large subunit of RNAPII (RPB1) and promote its transition into a productive elongation complex, increasing mRNAs synthesis mediated by RNAPII [1,4,5,6,7,8,9].

The function of P-TEFb is also regulated by the 7SK small nuclear ribonucleoprotein complex (7SK snRNP) composed of 7SK RNA, HEXIM1, LARP7, and MePCE [10,11]. HEXIM1 contains three major functional domains: an RNA-binding domain (RBD) containing an arginine-rich motif (ARM), an inhibitory domain (ID) that inhibits CDK9 activity and a CytT1-binding domain (TBD). HEXIM1 binds to 7SK RNA via RBD and to P-TEFb via TBD and ID, thereby constructing the P-TEFb:HEXIM1:7SK complex. ID of HEXIM1, which features a PYNT motif (202Pro-203Tyr-204Asn-205Thr), suppresses the CDK activity of P-TEFb by masking the substrate-binding site of CDK9 [12,13,14,15]. In addition, the TBD domain inhibits CDK9 activity synergistically with ID [16].

Human immunodeficiency virus-1 (HIV-1) exploits host cellular factors to transcribe its genes. A major contributor to HIV transcription is the well-known viral trans-activator Tat, which consists of two crucial domains: an RBD containing ARM and a transactivation domain (AD) binding to the transcriptional machinery of host cells [12,17]. The ARM of Tat, which is nearly identical in amino acid sequence to the HEXIM1′s ARM, can bind to 7SK RNA and releases P-TEFb from 7SK snRNP. It can also bind to the bulge region of the transactivation response element (TAR). Tat also binds to P-TEFb components and TAR via AD. The CycT1 binds to AD and the Tat-TAR recognition motif of CycT1 makes Tat and CycT1 bind to the central stem-loop structure of TAR cooperatively [12,18,19]. To further activate transcriptional elongation, Tat hijacks P-TEFb from HEXIM1 and transfers P-TEFb to the viral genome through the Tat–TAR interaction to form P-TEFb:Tat:TAR complex [18,19]. After escape from HEXIM1, CDK9 phosphorylates RNAPII, thereby enhancing transcriptional elongation initiated from the HIV-1 long terminal repeat (LTR) region [20,21,22].

Previously, a chimeric fusion protein composed of the ID of HEXIM1 and AD of Tat was synthesized and its antiviral activity was examined. This transcriptional inhibitor, named HEXIM1-Tat (HT), can effectively repress HIV-1 transcription [23].

HIV-1 nucleocapsid (NC), an RBD of Gag, has two CCHC zinc finger motifs with high affinity for nucleic acids [24,25,26,27]. NC contributes to the formation of the viral core and packages two copies of the full-length RNA genome into each viral particle by selectively binding to a so-called packaging signal sequence located in the leader region of HIV-1 genomic RNA (gRNA).

HIV-1 leader RNA has four major stem-loop structures (SL1, SL2, SL3, and SL4) that contribute to gRNA packaging. SL1 includes the dimer initiation signal, which forms a gRNA–gRNA dimer. SL2 contains the major splice donor [28]. The Psi site (Ψ) present in SL3 is the most conserved region in the HIV-1 genome and binds with NC with high affinity to promote RNA packaging [29,30,31,32,33,34,35,36]. In this case, NC can function as a chaperone protein that allows HIV full-length RNA to have an efficient structure for packaging [37,38,39]. We reasoned that NC could act as an antiviral agent by binding to viral RNA. Thus, this study examined whether the newly synthesized fusion proteins NC-HEXIM1-Tat (NHT) and HEXIM1-Tat-NC (HTN) have better anti-HIV effects. We found that both proteins exerted an enhanced antiviral effect by acting on multiple HIV life cycle steps and effectively inhibited the production of progeny viruses.

## 2. Results

### 2.1. The Chimeric Proteins NHT and HTN Have Better Antiviral Effects than HT

HT is an HIV-1 transcriptional repressor designed to possess both P-TEFb ID of HEXIM1 and AD of Tat [23]. Using it as a template, we fused NC with HT to generate NHT and HTN (Figure 1).

To determine whether NHT, HTN, and HT effectively inhibit HIV-1 replication, each chimeric protein was co-transfected with NL4-3EGFP and DsRed into HEK293T cells. Overexpression of multiple proteins might repress overall transcription and/or translation, leading to decreased virus production without any specific effect of the chimeric proteins on HIV-1. To avoid this error, we used a red fluorescent protein (RFP) expression vector (DsRed) as an internal control to monitor the transfection efficiency and determined GAPDH expression levels as a housekeeping protein. Each chimeric protein significantly reduced p24 expression in cell lysates (Figure 2a,b). p24 expression was decreased by 86.5% by NHT, 74.6% by HTN and 74.5% by HT. Additionally, NHT, HTN and HT suppressed NL43-derived GFP expression by 49.4, 22.1, and 39.2%, respectively (Figure 2c), and they also reduced the amount of virus released into the viral supernatant by 93.6, 68.0, and 44.3%, respectively (Figure 2d). Notably, NHT and HTN reduced viral release by 2.1- and 1.5-fold, respectively, versus the effects of HT.

Next, we infected MT4 cells with viral supernatant obtained from HEK293T cells co-transfected with NL4-3EGFP and NHT, HTN, or HT. Using the same volume of viral supernatant, the number of virus-infected cells was decreased by 59.9% in the NHT group, 59.7% in the HTN group and 17.3% in the HT group (Figure 2e,f). These results illustrate that the NC-fused chimeric proteins exhibit stronger anti-HIV effects than HT and suggested that NC-fused chimeric proteins could inhibit HIV-1 production by different mechanisms than HT. Therefore, we compared the mechanisms of action of NHT, HTN, and HT.

### 2.2. NHT, HTN, and HT Inhibit the Transcription of the HIV-1 Gene in a Tat-Dependent Manner

Experiments were undertaken to verify the effect of each chimeric protein on transcription. The HIV-1 proviral DNA used for the transfection assay carries a 5′ LTR containing the U3RU5 sequence. It functions as a promoter for binding to RNAPII and transcription of HIV-1 leader RNA is initiated at the R site, which contains the TAR sequence [40,41]. Therefore, we performed a reporter gene assay with firefly luciferase (Fluc) expression and either U3RU5 or RU5 sequences as the promoter. U3RU5-Fluc and U3RU5Ψ-Fluc vectors were expressed consistently with the chimeric proteins (Figure 3a,b). RFP expression vector was used to monitor the transfection efficiency and Fluc intensities were normalized to that of RFP.

We also used the tetracycline-inducible system to regulate the reporter gene tightly and ensure measuring precisely better the effect of the chimeric protein on transcription per se (Figure 3c,d). This could help ensure that the cellular translation machinery was not overwhelmed by the expression of several proteins and enabled us to measure reporter gene expression more accurately. In the absence of Tat, none of the chimeric proteins caused significant changes in the activity of the LTR promoter with all reporter vectors. Reporter gene expression was escalated in the presence of Tat by the transactivation effect. Conversely, the increased reporter gene expression induced by Tat was significantly reduced when NHT, HTN, or HT was co-expressed. When the promoter featured the U3RU5 sequence, Fluc expression was reduced by 80.0% by NHT, 54.0% by HTN, and 87.7% by HT. When U3RU5Ψ was used, Fluc expression was reduced by 87.4% by NHT, 91.5% by HTN, and 94.8% by HT (Figure 3b). When the tetracycline-inducible system was used in T-Rex 293 cells with the RU5 sequence as the promoter, Fluc expression was decreased by 55.1% by NHT, 37.0% by HTN, and 58.7% by HT (Figure 2c). Using RU5Ψ, Fluc expression was decreased by 74.0% by NHT, 53.8% by HTN, and 72.7% by HT (Figure 3d). However, there was no notable difference in the transcription-inhibitory effect based on the presence of the Ψ site in the reporter gene vector.

Then, we further confirmed whether Gag mRNA expression was noticeably altered when NHT, HTN, or HT was co-transfected with the proviral vector NL4-3EGFP into HEK293T cells. The results illustrated that the level of transcription products decreased by 32.8%, 29.9%, and 29.2% by NHT, HTN, and HT transfections, respectively (Figure 3e, left graph). Afterward, we additionally overexpressed Tat under the same conditions. If the chimeric proteins compete with Tat, the additional expression of Tat should reduce the inhibitory effect on viral transcription. As we expected, the inhibitory effect of each chimeric protein was offset by Tat, and the level of the transcription product level was restored when Tat was overexpressed (Figure 3e, right graph).

### 2.3. NTH, HTN, and HT Interact with and Hijack the Host Cell Factor P-TEFb and Viral RNA by Competing with Tat

To further examine the mechanism by which chimeric proteins compete with Tat, we investigated protein–protein and protein–RNA interactions. First, we confirmed whether NHT, HTN, or HT could interact with the P-TEFb subunits CDK9 and CycT1 in co-immunoprecipitation (co-IP) experiments. HEK293T cells were co-transfected with each Myc-tagged chimeric protein, Flag-tagged Tat, and NL4-3EGFP, and the cell lysate was pulled down with Myc antibody to detect chimeric protein-bound P-TEFb. NHT, HTN, and HT concentration-dependently interacted with CDK9 and CycT1 (Figure 4a). However, the amounts of CDK9 and CycT1 bound to HIV-1 Tat decreased as the expression of the chimeric proteins increased when Tat was pulled down using a Flag antibody (Figure 4b). Therefore, all chimeric proteins could bind P-TEFb competitively with Tat.

Next, we confirmed the interaction between the chimeric proteins and viral RNA (vRNA) using RNA immunoprecipitation (RIP). The affinity of each chimeric protein for TAR was measured by pulling down the chimeric protein with Myc antibody and quantifying the amount of TAR RNA via reverse transcription-polymerase chain reaction (RT-PCR). The interaction between each chimeric protein and TAR increased in a concentration-dependent manner (Figure 5a). Conversely, NHT, HTN, and HT decreased the Tat–TAR interaction by 32.5%, 16.6%, and 21.1%, respectively. Thus, each chimeric protein interacted with TAR and competitively inhibited the Tat–TAR interaction.

The reporter vectors U3RU5-EGFP, U3RU5Ψ-EGFP, and Ψ-EGFP were examined and observed under the same condition as NL4-3EGFP. Both U3RU5-EGFP and U3RU5Ψ-EGFP enhanced the protein–TAR interaction in a chimeric protein-dependent manner. Conversely, U3RU5-EGFP tended to decrease the Tat–TAR interaction when each chimeric protein was co-expressed (Figure 5b). However, NHT significantly reduced the Tat–TAR interaction by 26.5% in the presence of U3RU5Ψ-EGFP (Figure 5c). Additionally, NTH and HTN, but not HT, exhibited high affinity for Ψ (Figure 5d). Therefore, we expected that the effective interaction of the NC-fused chimeric protein with the Ψ site could be the reason that NHT and HTN outperformed HT’s antiviral effect.

### 2.4. NTH and HTN Interrupt vRNA Packaging and Reduce the Infectivity of the Progeny Virus

Because NC-fused chimeric proteins effectively interacted with the Ψ site, we supposed that NHT and HTN could affect vRNA packaging. Therefore, we measured the amount of vRNA genomes packed into the released virus particles. The number of virus particles was quantitated by using HIV-1 p24 enzyme-linked immunosorbent assay (ELISA), and then vRNA extracted from the quantitated virus particles was subjected to quantitative RT-PCR using Gag complementary primers. Each value was normalized to the p24 level measured by p24 ELISA. The results illustrated that NHT and HTN decreased vRNA incorporation into virus particles by 47.7% and 27.1%, respectively, whereas HT had no effect. Compared to HT, NHT and HTN showed reductions in RNA incorporation up to 34.5% and 13.9%, respectively (Figure 6a). Additionally, NHT and HTN markedly decreased the infectivity of released viruses by 63.6% and 42.1%, respectively. When compared to HT, the magnitude of infectivity reduction would be 46.2% and 24.6%, respectively (Figure 6b,c). Altogether, our data demonstrate that NTH and HTN effectively inhibited HIV-1 replication by interfering with both viral transcription and packaging.

## 3. Discussion

A previous study demonstrated that HT, which is a composite of P-TEFb ID of HEXIM1 and AD of Tat, inhibited the transcription of the HIV-1 genome [19,23]. We fused the NC domain to HT to further enhance its antiviral effect. Therefore, we constructed NHT and HTN and confirmed their anti-viral effects. NC strongly binds to viral RNA via two CCHC zinc finger motifs [24,25], so we expected that NC could have HIV-1-specific action by helping the chimeric proteins bind specifically to vRNA and reducing the off-target effect.

Notably, NHT and HTN outperformed HT in inhibiting intracellular viral protein synthesis and virus release. HT exerted only a minor effect on HIV-1 replication when MT4 cells were infected with viruses released from HEK293T cells in the transfection assay. Thus, we further investigated how NHT and HTN obtained additional antiviral effects. First, we verified whether NC-fused chimeric proteins have an enhanced viral transcriptional inhibition effect compared to HT. We examined the effect of NC-fused chimeric proteins on the level of LTR-mediated transcription since HT has been designed to inhibit viral transcription by competing with Tat [23]. In the presence of Tat, NHT, HTN, and HT effectively suppressed Tat-stimulated LTR promoter activity. Thus, each chimeric protein effectively inhibited HIV-1 transcription as designed.

Because NC has high affinity for the Ψ site of HIV-1 RNA [29,30,31,32,33,34,35,36], we used the promoter including the Ψ sequence, to confirm the influence of the NC–Ψ interaction on transcriptional activity. However, we found no evidence that the NC–Ψ interaction directly affected the activity of the LTR promoter. In addition, NHT and HTN did not surpass the transcriptional-inhibitory effect of HT. Thus, this suggests that the Ψ site is not involved any further in the NHT-and HTN-mediated inhibition of HIV-1 transcription.

The regulation of HIV-1 transcription by the chimeric proteins was also confirmed by reducing intracellular HIV-1 transcripts. Unlike in the reporter gene assay using partial LTR sequences of HIV-1, NHT most effectively reduced Gag mRNA synthesis in full-length pro-viral DNA. NTH could possibly repress viral transcription via a different mechanism than regulating the activity of the LTR promoter. NC can change the tertiary structure of vRNA and control the pressure to cause transcription or packaging [37,38,39]. NC has also been reported to regulate the activity of HIV-1 reverse transcription (RT) [42,43,44], and an NC mimic impaired RT and inhibited viral replication [45]. Therefore, the additional repression of proviral DNA transcription by NHT might be due to a decrease in the pressure on transcription or a change in the regulation of RT activity, and further studies are required to clarify this aspect. Additionally, Tat overexpression restored viral transcription to a level similar to that in the control group. Therefore, we suggested that each chimeric protein could inhibit HIV-1 transcription in a Tat-competitive manner.

Next, we investigated whether NHT, HTN, and HT bind P-TEFb through AD of Tat and TBD of HEXIM1. In line with their design, all chimeric proteins exhibited high affinity for CycT1 and CDK9 and competitively hijacked P-TEFb from Tat. Moreover, NHT, HTN, and HT have ARM of HEXIM1, which is almost identical to that of Tat [12]. Thus, we verified whether each chimeric protein binds to TAR RNA and competes with Tat for TAR binding. As reported in a previous study on HT [23], each chimeric protein bound to TAR effectively and competitively inhibited the Tat–TAR interaction by hijacking TAR RNA. Therefore, we suggested that NHT, HTN, and HT could interact with vRNA TAR and P-TEFb effectively and thus act as Tat antagonists.

When the HIV-1 gene is transcribed, Tat forms a P-TEFb:Tat:TAR complex mimicking the 7SK snRNP complex (P-TEFb:HEXIM1:7SK) and recruits P-TEFb to the HIV-1 promoter [12]. Activated P-TEFb phosphorylates RPB1, a large subunit of RNAPII, causing RNAPII to initiate transcriptional elongation at the HIV-1 promoter site [20,21,22]. If one of the chimeric proteins is present in this transcription process, then the chimeric proteins can induce the formation of the P-TEFb:chimeric protein:TAR complex. The activity of P-TEFb bound to the chimeric protein can be inhibited by ID and TBD. Therefore, chimeric proteins could selectively bind to viral transcripts and inhibit transcription elongation.

Taken together, both NHT and HTN could inhibit HIV-1 transcription by retaining the function of HT, and the NC sequence fused to NHT or HTN did not alter the original mechanism of HT. Thus, we investigated the mechanism of NC-fused chimeric proteins that act independently of the transcriptional inhibitory effect. Interestingly, we found that the affinity of the chimeric proteins for U3RU5Ψ RNA was higher than that of HT, in contrast to their affinity for U3RU5 RNA. NHT and HTN exhibited high affinity for the Ψ site of viral RNA, whereas HT barely interacted with Ψ-EGFP. Additionally, NHT and HTN inhibited vRNA incorporation, thereby leading to the secretion of non-infectious virus particles and repressing the infectivity of HIV-1 progeny viruses.

The interaction of NC with the Ψ site of viral RNA facilitates vRNA packaging by forming the viral-specific ribonucleoprotein complex [33,35]. Thus, we suggested that NHT and HTN block regular RNA packaging by occupying vRNA via interaction with the Ψ site, thereby reducing the infectivity of HIV-1.

By using the characteristics of NC in this study, we developed novel antiviral peptides that outperform HT. The NC-fused chimeric proteins could maintain the competitive transcriptional inhibition of Tat and improve the inhibitory effect on HIV-1 replication by providing an additional interaction with the Ψ site of the vRNA genome. They could selectively bind to the viral genome as a competitor and bolster the inhibitory effect on viral replication. Therefore, NC-fused chimeric proteins can be exploited as novel molecules of HIV-1 that inhibit both vRNA packaging and infectivity as well as inhibition of viral transcription. Furthermore, NHT and HTN can inhibit HIV-1 proliferation effectively since they inhibit both virus production and decrease the infectivity of the virus produced. 

Additionally, with the rapid development of various peptide delivery routes like oral or transdermal, the convenience of administering peptide therapeutics is rapidly improving [46,47]. The current treatment for HIV-1 is mainly combination antiretroviral therapy (cART), which uses several types of drugs that target different stages of the HIV-1 life cycle to suppress the virus proliferation. However, curing HIV-1 with cART is difficult, which endangers a high risk of drug resistance and side effects owing to the long-term use of the drug [48]. Therefore, we highlighted the possibility of developing anti-HIV molecules, which regulate multiple stages of the HIV-1 life cycle and can replace conventional cART. This type of engineering would be expected to apply to the development of new therapeutics for HIV-1 as well as other viruses.

## 4. Materials and Methods

### 4.1. Cell Culture and Transfection

HEK293T (ATCC, VA, USA) and T-Rex 293 (Invitrogen, MA, USA) cells were cultured in Dulbecco’s modified Eagle’s medium (Hyclone, Cat. #SH30243.01) supplemented with 10% fetal bovine serum (Hyclone, Cat. #SH30919.03) and 1% penicillin/streptomycin (Gibco, Cat. #15240-062) at 37 °C and 5% CO_2_. MT4 (Invitrogen) cells were maintained in the same manner except for the use of RPMI 1640 medium. For transient transfection, 1 × 10^5^ HEK293T or T-Rex 293 cells were plated in each well of 24-well plates. We transfected 80%–90% confluent cells with the plasmids using Lipofector-pMax (Aptabio, Gyeonggi, Korea) or jetPEI (Polyplus-Transfection, Illkirch-Graffenstaden, France) according to the manufacturer’s instructions.

### 4.2. Antibodies

Anti-Myc (Cat. #sc-40), anti-p24 (Cat. #sc-69728), anti-GFP (Cat. #sc-9996), anti-GAPDH (Cat. #sc-47724), anti-CycT1 (Cat. #sc-271348), and anti-CDK9 (Cat. #sc-13130) antibodies were obtained from Santa Cruz Biotechnology. Anti-FLAG (Cat. #F3165) antibody was purchased from Sigma-Aldrich. Anti-phospho-Rpb1 CTD (Ser2) (Cat. #13499) antibody was purchased from Cell Signaling Technology.

### 4.3. Preparation of the HIV-1 Virus Stock

A recombinant EGFP-expressing HIV-1 virus was generated by transfecting HEK293T cells with a pNL4-3EGFP pro-viral plasmid carrying an EGFP gene in place of the nef gene. We harvested the viral supernatant from transfected HEK293T cells and separated the sludge by centrifugation at 10,000× *g* for 2 min and stored it at −80 °C before use.

### 4.4. HIV-1 p24 ELISA

ELISA for HIV-1 p24 was performed according to the manufacturer’s instructions (XpressBio, Cat. #XB-1000). Briefly, we serially diluted 10 μL of each harvested viral supernatant 10-fold in cultured medium. Then, 20 μL of lysis buffer and 200 μL of the diluted specimen were added to each well of the microtitration plate. After incubating the mixture for 1 h at 37 °C, we washed the plate four times with 350 μL of wash buffer. Then, 100 μL of detector antibody were added, and the mixture was incubated for 1 h at 37 °C. We washed the plate again and dispensed 100 μL of substrate solution into each well. After incubation for 30 min at room temperature, 100 μL of stop solution were added to each well, and the absorbance was measured at 450 nm. We used positive controls to quantify the absolute amount of p24 antigen from each virus according to the manufacturer’s instructions.

### 4.5. MT4 Cell Infection

The viral supernatants obtained from HEK293T cells were added to 1.4 × 10^5^ MT4 cells and incubated at 37 °C for 72 h. Next, the infected cells were observed by fluorescence microscopy (Axiovert 200, Zeiss, Germany). The FACS Canto instrument (BD Biosciences, CA, USA) was used to quantify GFP-positive MT4 cells.

### 4.6. Reporter Gene Assay

For the reporter gene assay, HEK293T cells were transfected with the reporter gene vector pGL3-U3RU5-Fluc, pGL3-U3RU5Ψ-Fluc, pCNA4-TetO-RU5-Fluc, or pCNA4-TetO-RU5Ψ-Fluc together with NTH, HTN, or HT (Table 1). The Fluc activity was measured according to the manufacturer’s instructions (Promega, Cat. #E1500). Briefly, cells were harvested and lysed after 24 h of incubation with passive lysis buffer and centrifuged briefly. The supernatant was transferred to a new tube and incubated with Luciferase Assay Reagent. The intensity of Fluc was measured using a Fluorometer INFINITE 200 (Tecan, Grödig, Austria). The data were normalized by the RFP level.

### 4.7. Reverse Transcription Polymerase Chain Reaction (RT-PCR)

For conventional RT-PCR, we incubated reaction mixtures containing extracted RNA, 2 U of DNase I, 5 mM MgCl_2_, and DEPC-treated water at room temperature for 1 h. To inactivate DNase I, EDTA was added at a final concentration of 5 mM, and the mixtures were incubated at 70 °C for 15 min. cDNA amplification was performed using M-MLV RT master mix (ELPIS Biotech, Cat. #EBT-1511) according to the manufacturer’s instructions. The PCR products were separated on a 1% agarose gel at 100 V by electrophoresis. The relative amount of each band was analyzed using Image Lab^TM^ Software (Bio-Rad, Hercules, CA, USA).

### 4.8. RNA Immunoprecipitation

RIP was performed using polysome lysis buffer (PLB) [49]. HEK293T cells co-transfected with pNL4-3EGFP and NHT, HTN, or HT were harvested in PLB containing 100 U/mL RNasin RNase inhibitor (Promega, Cat. #N2115), 2 mM vanadyl ribonucleoside complexes solution (Sigma-Aldrich, Cat. #94742), and a protease inhibitor cocktail (Roche, Cat. #04693132001). Lysates were precleared for 2 h at 4 °C using protein A/G PLUS agarose beads (Santa Cruz Biotechnology, Cat. #sc-2003) washed with PLB twice, and the precleared lysates were incubated for 4 h at 4 °C with 2 μg of the indicated primary antibody. Protein A/G agarose beads were added, followed by overnight incubation in a rotator at 4 °C. After that, protein A/G agarose beads were washed four times with PLB for 5 min on a 4 °C rotator, and then RNA was extracted using RNAiso Plus (TaKaRa, Cat. #9109) according to the manufacturer’s manual. The extracted RNA was used for RT-PCR as previously described.

### 4.9. Co-Immunoprecipitation

Co-IP was performed in the same manner as described for RIP up to the addition of protein A/G agarose beads to lysates. However, phosphate-buffered saline was used as a washing buffer. Washed protein A/G agarose beads resuspended in 2X Laemmli buffer and incubated for 5 min at 95 °C. The supernatants were subjected to Western blot analysis.

### 4.10. Statistical Analysis

All values are expressed with the standard error of the mean. We used one-way ANOVA to analyze the quantitative data. Dunnett’s test was used to compare each treatment group with the control group. *p*-values less than 0.05 were considered statistically significant.

## 5. Conclusions

The novel chimeric proteins NHT and HTN inhibited HIV-1 proliferation effectively. The results illustrated that NHT and HTN can act as Tat antagonists by binding to TAR RNA and P-TEFb and reduce the infectivity of the progeny virus by inhibiting the vRNA packaging, thereby highlighting their potential for development as effective treatments that regulate multiple steps in the HIV-1 life cycle.

## Figures and Tables

**Figure 1 ijms-23-12340-f001:**
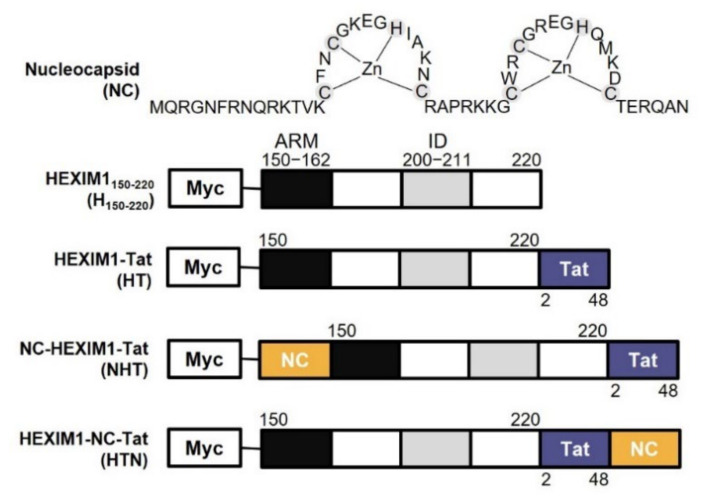
The scheme of HEXIM1 derived chimeric proteins. The structure of the NC zinc finger motifs and functional domains of HEXIM1 and Tat, namely ARM (black box, residues 150–162), which binds RNA, ID (gray box, residues 200–211), which inhibits CDK9 through a PYNT motif, and AD (blue box, residues 2–48), which binds to P-TEFb. NC (yellow box, residues 1–55) was inserted in front of or behind HT. All chimeric proteins were tagged with Myc at the N-terminus. The size of each box does not reflect the actual size of the domain. NC, nucleocapsid; ARM, arginine-rich motif; ID, inhibitory domain; AD, transactivation domain; P-TEFb, positive transcription elongation factor b; HT, HEXIM1–Tat.

**Figure 2 ijms-23-12340-f002:**
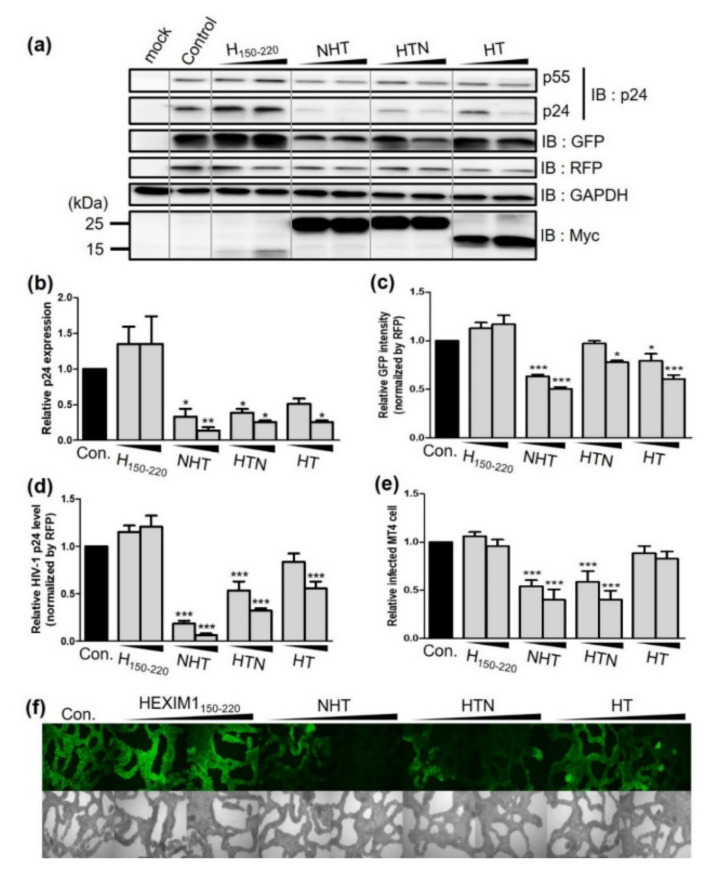
Effect of HEXIM1 derived chimeric proteins on the virus production in HEK293T cells. (**a**) Western blotting results using cell lysates. Upper panel: expression of viral proteins, RFP transfection control and GAPDH as the housekeeping gene. Bottom panel: expression of H_150-220_ and its derivates. (**b**) Intensity of the p24 band in western blotting (*n* = 3). (**c**) Intensity of the GFP signal derived from the proviral vector in the cell lysate (*n* = 5). (**d**) Supernatant viral titer quantified by p24 ELISA and normalized by the RFP level (*n* = 5). (**e**,**f**) Viral supernatant harvested from transfected HEK293T cells was used to transduce 1.4 × 10^5^ MT4 cells. MT4 cells were infected with the same volume of viral supernatant. Detection of GFP-positive infected cells by FACS (**e**) and fluorescence microscopy (**f**). All the microscopy images were at 100× magnification. Statistical significance was assessed using one-way analysis of variance (ANOVA; * *p* < 0.05, ** *p* < 0.01, *** *p* < 0.005). RFP, red fluorescent protein; GFP, green fluorescent protein; GAPDH, glyceraldehyde-3-phosphate dehydrogenase; ELISA, enzyme-linked immunosorbent assay; FACS, fluorescence-activated cell sorting.

**Figure 3 ijms-23-12340-f003:**
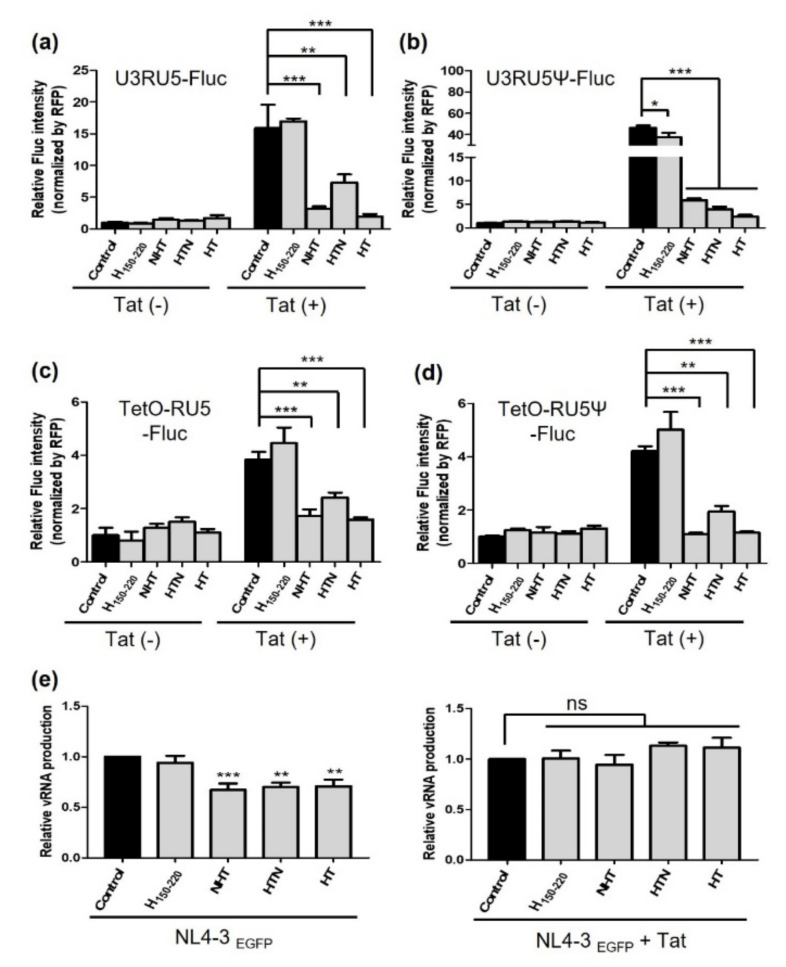
NHT, HTN and HT inhibited viral transcription in the presence of Tat. (**a**,**b**) HEK293T cells transfected with the empty vector; Myc-HEXIM1150-220; or Myc-tagged NHT, HTN, or HT with the reporter vector and RFP. Additionally, an expression vector for flag-tagged Tat was co-transfected a double the amount of the chimeric protein to confirm the effect of Tat. Twenty-four hours after transfection, cells were lysed in passive lysis buffer containing proteinase inhibitor cocktail and Fluc activity was measured (*n* = 3). (**c**,**d**) Vectors with the TetO site were designed to regulate the transcription of the reporter gene using a tetracycline-inducible system. T-Rex 293 cells were also co-transfected in the same manner as HEK293T cells. Eighteen hours after transfection, reporter gene expression was induced with 1 μg/mL tetracycline, cells were lysed and Fluc activity was detected 24 h after induction (*n* = 3). U3RU5-Fluc (**a**), U3RU5Ψ-Fluc (**b**), TetO-RU5-Fluc (**c**), and TetO-RU5Ψ-Fluc (**d**) were used as reporter gene vectors. (**e**) HEK293T cells were co-transfected with Myc-tagged NHT, HTN, or HT and NL4-3EGFP with or without Tat. Twenty-four hours after transfection, RT-PCR was performed using primers complementary to Gag. Statistical significance was assessed using one-way ANOVA and Bonferroni’s multiple comparison test (* *p* < 0.05, ** *p* < 0.01, *** *p* < 0.005). NHT, NC-HEXIM1-Tat; HTN, HEXIM1-Tat-NC; Fluc, firefly luciferase; TetO, tetracycline operator; RT-PCR, reverse transcription-polymerase chain reaction.

**Figure 4 ijms-23-12340-f004:**
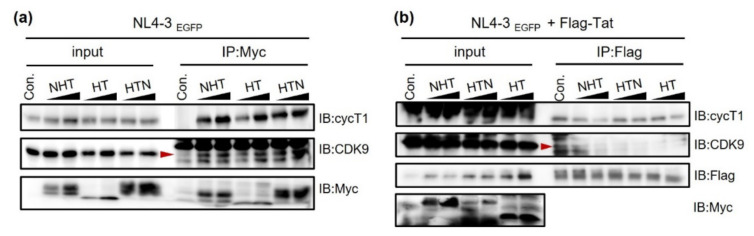
NHT, HTN, and HT interact with the transcriptional machinery. HEK293T cells were co-transfected with Myc-tagged NHT, HTN, or HT and NL4-3EGFP with or without Flag-Tat. Twenty-four hours after transfection, cells were lysed in passive lysis buffer. (**a**) The interactions of Myc-tagged chimeric proteins with the P-TEFb components CycT1 and CDK9. Each Myc-tagged protein in the cell lysate was pulled down by Myc antibody, and the interaction with P-TEFb was verified. (**b**) The interactions of the Flag-Tat with CycT1 and CDK9. The overexpressed Flag-Tat in the cell lysate was pulled down by Flag antibody, and its interaction with P-TEFb was verified. The red arrows indicate the CDK9 band.

**Figure 5 ijms-23-12340-f005:**
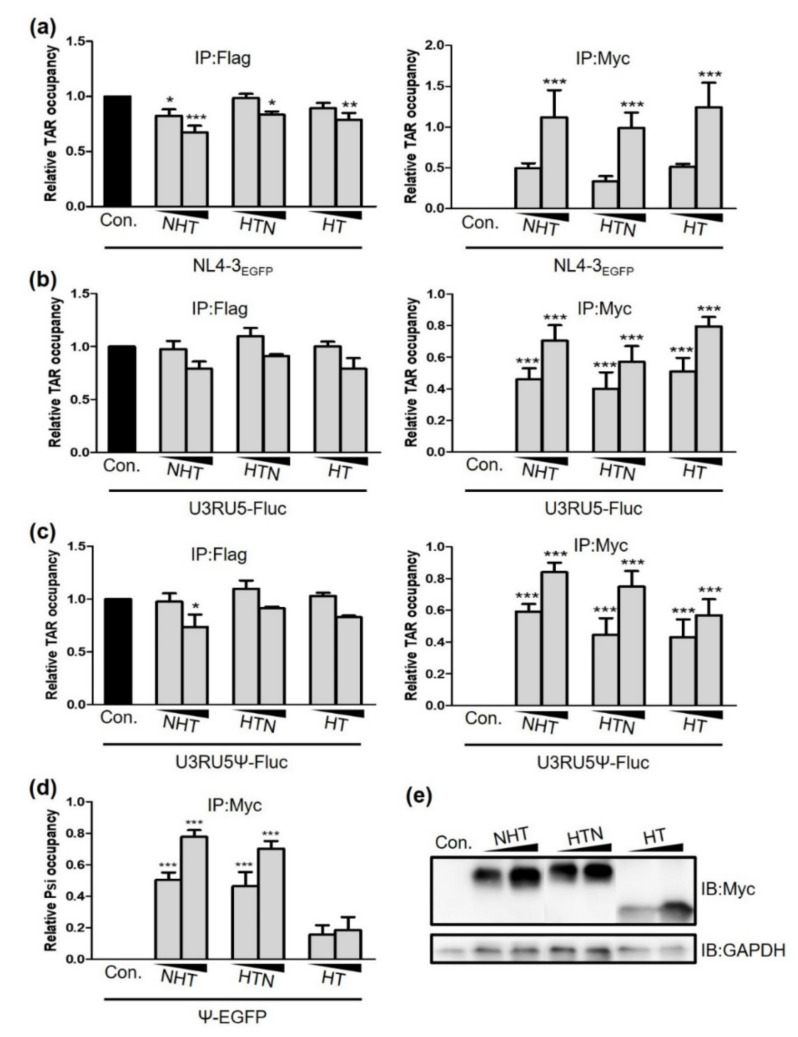
NHT, HTN, and HT interact with viral RNA. HEK293T cells were transfected with the empty vector; Myc-tagged NHT, HTN, or HT, or Flag-tagged Tat with NL4-3EGFP (**a**), U3RU5-Fluc (**b**), U3RU5Ψ-Fluc (**c**) or Ψ-EGFP (**d**). Twenty-four hours after transfection, cells were lysed in passive lysis buffer, and each cell lysate was pulled down by Myc or Flag antibody. RT-PCR was performed using primers complementary to TAR or Ψ to measure the amount of viral RNA bound to each chimeric protein or Tat. In the “IP: Myc” graphs, each value was obtained by subtracting the amount of vRNA bound non-specifically in the control group from the raw data. (**e**) Expression of NHT, HTN, and HT. Statistical significance was assessed using one-way ANOVA (* *p* < 0.05, ** *p* < 0.01, *** *p* < 0.005). TAR, transactivation response element.

**Figure 6 ijms-23-12340-f006:**
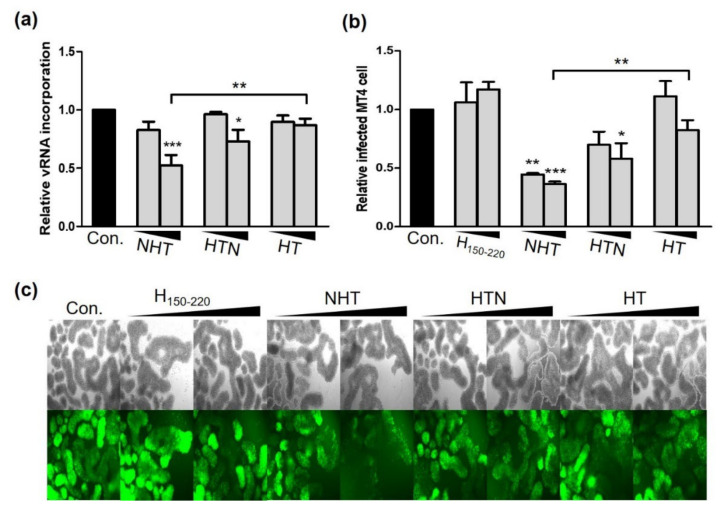
NHT and HTN interrupted vRNA packaging and restrained the infectivity of the progeny virus. (**a**) The amount of vRNA incorporated into the released virus particles was measured by RT-PCR. (**b**,**c**) The viral supernatant harvested from transfected HEK293T cells was used to transduce 1.4 × 10^5^ MT4 cells. MT4 cells were infected with viral supernatant containing the same amount of p24 particles as quantified by ELISA. Detection of GFP-positive infected cells by FACS (**b**) and fluorescence microscopy (**c**). All the microscopy images were at 100× magnification. Statistical significance was assessed using one-way ANOVA (* *p* < 0.05, ** *p* < 0.01, *** *p* < 0.005).

**Table 1 ijms-23-12340-t001:** The LTR promoter sequences of reporter vectors.

LTR Site	Sequence
U3	tggaagggctaatttggtcccaaaaaagacaagagatccttgatctgtggatctaccacacacaaggctacttccctgattggcagaactacacaccagggccagggatcagatatccactgacctttggatggtgcttcaagttagtaccagttgaaccagagcaagtagaagaggccaaataaggagagaagaacagcttgttacaccctatgagccagcatgggatggaggacccggagggagaagtattagtgtggaagtttgacagcctcctagcatttcgtcacatggcccgagagctgcatccggagtactacaaagactgctgacatcgagctttctacaagggactttccgctggggactttccagggaggtgtggcctgggcgggactggggagtggcgagccctcagatgctacatataagcagctgctttttgcctgtact
R	gggtctctctggttagaccagatctgagcctgggagctctctggctaactagggaacccactgcttaagcctcaataaagcttgccttgagtgctc
U5	aaagtagtgtgtgcccgtctgttgtgtgactctggtaactagagatccctcagacccttttagtcagtgtggaaaatctctagca
Ψ	gcaggactcggcttgctgaagcgcgcacggcaagaggcgaggggcggcgactggtgagtacgccaaaaattttgactagcggaggctagaaggagagagatgggtgcgagagcgtcggtattaagcg

## Data Availability

Not applicable.

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
