# Peer review of "Identification of Novel Nucleocapsid Chimeric Proteins Inhibiting HIV-1 Replication"

_ijms, 2022, doi:10.3390/ijms232012340_

Round 1
Reviewer 1 Report
This study investigated the effect of two artificial fusion proteins on HIV-1 expression and virus production. The chimeric proteins in question are based on a Hexim1-Tat (HT) fusion, to which nucleocapsid (NC) has been added either to the C- or N-terminus to create HTN and NHT, respectively. Expression of these chimeric proteins reduce Gag expression in transfected cells, which causes reduced virion production. Reporter and co-IP assays show that transcription from the LTR is downregulated and that HTN and NHT compete with Tat to bind PTEFb. It is reported that the chimeric proteins interfere with viral RNA genome packaging.
If you overexpress (OE) a protein containing the activation domain of Tat and this protein competes with regular full-length Tat, is this really a surprising result? Is this specific for HIV?
L57 NC is described as an RBD of HIV, but it’ a domain of Gag. HIV is a virus, no domains.
If the hybrid proteins are myc tagged, figure 1 depictions should reflect that.
Section 2.1 The two fusion proteins (NHT, HTN) already resulted in less p24 expression in cells, therefore less virus release. So it’s not so surprising that using the same volume of virus results in less infected cells. What would be more important is to show what happens if you use the p24-normalized amounts of virus, to see whether there is a defect in infectivity of the particles that are produced.
It seems as though the majority of Gag in Fig 2a is not processed properly, even in the absence of any fusion protein being expressed. It’s mostly Pr55, rather than p24. The reasons are not clear to me.
Another point is that a lot of proteins are being overexpressed in cells, thus the reduced expression of p24 might reflect an overwhelmed cellular/translational state rather than a specific inhibition of Gag production. A control showing the expression of another irrelevant protein (at similar levels) would be needed to ensure this is not the case. HEXIM1-150-220 construct does not seem to be expressed based on the myc western blot. Not a surprise that it doesn’t change much from the control.
The legend of Figure 2 talks about colored boxes, but these do not exist in Figure 2. It is confusing. There is talk of a “g” section, for which there is no figure. The explanations are shifted. From the microscopy pictures, it’s not possible to see any difference in terms of number or intensity of the GFP signal.
Figure 3; “RFP related Fluc intensity” - Probably Fluc that is normalized to RFP is meant? This should be stated better.
NHT or HTN do not show any difference in tat-mediated expression from the LTR, and that they don’t seem more efficient than HT. So the transcriptional differences do not fully explain why these two are better than HT.
It is unclear to me why the Tet-system is used for assessing transcriptional activation. They have Tat-mediated LTR transcriptional activation, fusion protein mediated repression of that, and another Tet-inducible promoter activation in the same system, which prevents one from understanding how a single system is affected by HTN or NHT expression. It is still entirely possible that the fusion proteins affect transcription in general.
L127. Statistical tests belong in the legend.
L176: HTN is written double.
2.3 In RIP experiments, Myc-tagged chimeric protein OE shows a clear dose-dependent increase in TAR occupancy, which makes sense. The more of the protein you have, the more it binds RNA. However, the competition the authors claim is trickier. The levels of Tat occupancy doesn’t really decrease much.
Fig 5 legend states Fig 4 in the beginning.
2.4. The title suddenly converts to past tense.
The authors already showed that p24 expression is lower, virus particles produced are lower in the presence of the chimeric proteins. If they do a qPCR for viral RNA in supes, of course they detect less RNA. These experiments would need to be normalized to p24 levels, if one wants to make the argument that viral RNA packaging is affected. In this case, it seems to me is that the gag production is affected. If the RT-PCR values were based on p24-normalized inputs, this should be stated clearly.
L244 - have they really looked at replication of HIV? It seems to me that the only experiments done were single-round.
From a basic research point of view, it’s interesting. But how would this become an effective treatment as claimed in L399?
Author Response
Thank you for your careful review.
We responded to your comments in word files.
Please check the uploaded file.

Reviewer 2 Report
In this manuscript, the authors described the fusion protein of HIV-1 NC with previously studied HEXIM1-Tat and its antiviral effects, as shown in reduced viral transcription or viral particle release. While the enhancement of viral suppression effect through addition of NC domain is interesting, the author would need to explain more clearly of the assays they performed.
My comments are as follow:
1) Can the authors make it more clear about the original hypothesis? Was it to utilize the high affinity of NC to nucleic acids, so the fused inhibitor could be brought to the viral RNA more efficiently?
2) The figure legend for figure 1 seemed to be misplaced under figure 2? There is also no mention of figure 2A in the main text. The viral release measurement shown in figure 2E/F don’t match with the main text either.
3) Assays performed to generate Figure 3, please specify the source and levels of Tat.
4) The low and high expression levels of NHT, HT, or HTN in figure 4 and figure 5 don’t always appear that way from the intensity of Myc. Maybe figure 4A is showing the differences, but figure 4B and 5E the Myc intensity looks saturated for what’s supposed to be low or high levels of fusion proteins.
5) Redundant phrases in line 215-219.
6) Figure 6, please explain how the authors discriminate between reducing viral release versus reducing the copies of viral RNA packed into each viral particle?
Author Response
Response to Reviewer 2 Comments
Point 1: Can the authors make it more clear about the original hypothesis? Was it to utilize the high affinity of NC to nucleic acids, so the fused inhibitor could be brought to the viral RNA more efficiently?
Response 1: Indeed, the point of view is what we are exactly trying to suggest. We have revised the manuscript to further clarify the point and make it easier for readers to understand.
Point 2: 1) The figure legend for figure 1 seemed to be misplaced under figure 2? There is also no mention of figure 2A in the main text. 2) The viral release measurement shown in figure 2E/F don’t match with the main text either. Redundant phrases in line 215-219.
Response 2: Thank you for your careful review. We have corrected what you pointed out.
Point 3: Assays performed to generate Figure 3, please specify the source and levels of Tat.
Response 3: We have specified the content that you pointed out on line 154.
Point 4: The low and high expression levels of NHT, HT, or HTN in figure 4 and figure 5 don’t always appear that way from the intensity of Myc. Maybe figure 4A is showing the differences, but figure 4B and 5E the Myc intensity looks saturated for what’s supposed to be low or high levels of fusion proteins.
Response 4: The images we presented for figure 4b and figure 5e were saturated as you said. We have replaced them with images showing the difference in protein expression levels.
Point 5: Figure 6, please explain how the authors discriminate between reducing viral release versus reducing the copies of viral RNA packed into each viral particle?
Response 5: We have added and explained further a description of the experiment in Figure 6a on line 237.
Round 2
Reviewer 1 Report
I don't have further comments.
Author Response
Thank you very much for your careful review.
Reviewer 2 Report
The authors corrected errors in main text and figure legends in the previous version, and replace blot images that reflect what they described. However, it is still unclear to me how the authors calculated the vRNA copy per viral particle in figure 6 --- more specifically, how the authors quantified the viral particles. Did the author perform a conventional viral titer assay, such as luciferase assay? If so, please include it in the materials and methods.
Additionally, for figure 6a and b, are the statistic analysis only comparing all experimental conditions to the "control"? Did the authors also perform statistic analysis between NHT/HTN to HT? Since the original thought was to show that NHT or HTN would further improve HT's antiviral effect, it seem to make a stronger point than just comparing with control.
